# Physiological and Transcriptomic Analysis Provides Insights into Low Nitrogen Stress in Foxtail Millet (*Setaria italica* L.)

**DOI:** 10.3390/ijms242216321

**Published:** 2023-11-14

**Authors:** Erying Chen, Ling Qin, Feifei Li, Yanbing Yang, Zhenyu Liu, Runfeng Wang, Xiao Yu, Jiahong Niu, Huawen Zhang, Hailian Wang, Bin Liu, Yanan Guan

**Affiliations:** 1Featured Crops Engineering Laboratory of Shandong Province, Crop Research Institute, Shandong Academy of Agricultural Sciences, Jinan 250100, China; chenerying_001@163.com (E.C.); qinling1021@163.com (L.Q.); lifeifei0951@hotmail.com (F.L.); ybyang_666@163.com (Y.Y.); liuzhenyu1@shandong.cn (Z.L.); lius.rw@outlook.com (R.W.); zhwws518@163.com (H.Z.); wanghailian11@163.com (H.W.); 18560216516@163.com (B.L.); 2College of Life Science, Shandong Normal University, Jinan 250014, China; yuxiao7896y@163.com (X.Y.); 2020020789@stu.sdnu.edu.cn (J.N.)

**Keywords:** foxtail millet, nitrogen use efficiency, varieties, physiological parameters, transcriptomics

## Abstract

Foxtail millet (*Setaria italica* (L.) P. Beauv) is an important food and forage crop that is well adapted to nutrient-poor soils. However, our understanding of how different LN-tolerant foxtail millet varieties adapt to long-term low nitrogen (LN) stress at the physiological and molecular levels remains limited. In this study, two foxtail millet varieties with contrasting LN tolerance properties were investigated through analyses of physiological parameters and transcriptomics. The physiological results indicate that JG20 (high tolerance to LN) exhibited superior biomass accumulation both in its shoots and roots, and higher nitrogen content, soluble sugar concentration, soluble protein concentration, zeatin concentration in shoot, and lower soluble sugar and soluble protein concentration in its roots compared to JG22 (sensitive to LN) under LN, this indicated that the LN-tolerant foxtail millet variety can allocate more functional substance to its shoots to sustain aboveground growth and maintain high root activity by utilizing low soluble sugar and protein under LN conditions. In the transcriptomics analysis, JG20 exhibited a greater number of differentially expressed genes (DEGs) compared to JG22 in both its shoots and roots in response to LN stress. These LN-responsive genes were enriched in glycolysis metabolism, photosynthesis, hormone metabolism, and nitrogen metabolism. Furthermore, in the shoots, the glutamine synthetase gene SiGS5, chlorophyll apoprotein of photosystem II gene SiPsbQ, ATP synthase subunit gene Sib, zeatin synthesis genes SiAHP1, and aldose 1-epimerase gene SiAEP, and, in the roots, the high-affinity nitrate transporter genes SiNRT2.3, SiNRT2.4, glutamate synthase gene SiGOGAT2, fructose-bisphosphate aldolase gene SiFBA5, were important genes involved in the LN tolerance of the foxtail millet variety. Hence, our study implies that the identified genes and metabolic pathways contribute valuable insights into the mechanisms underlying LN tolerance in foxtail millet.

## 1. Introduction

Nitrogen is an important and essential macronutrient for plant growth and development [1,2], and is part of all building blocks of life, including nucleic acids, amino acids, proteins, lipids, and metabolic products [3]. In recent decades, nitrogen fertilizer, as an important input factor for crop production, has played an important role in increasing crop yield [4]. However, the cost of nitrogen fertilizer accounts for a large proportion of agricultural input [5]. It has been reported that no more than 40% of the applied N is taken up by crops, and most of the applied N is retained in the soil and lost to the atmosphere, groundwater, and rivers through volatilization, leaching, and surface runoff [6]. The adverse side effects of nitrogen application have seriously damaged the environment [6,7,8]. It was reduced by a series of environmental policies, such as Nitrates Directive within the EU [9,10]. Reducing the nitrogen application and improving nitrogen use efficiency (NUE) is necessary for agricultural sustainability [11]. NUE comprises two components: nitrogen uptake efficiency (NUpE) and nitrogen utilization efficiency (NUtE) [12]. Numerous studies have shown that there exist significant genetic differences in NUE and its two components [13,14,15,16]. NUE refers to the amount of dry matter produced by plants absorbed per unit mass of nitrogen and results from the coordination of carbon and nitrogen metabolism [17]. Nitrogen metabolism includes N absorption [6,18], mainly including nitrate (NO_3_^−^) and ammonium (NH_4_^+^) under different soil conditions [6]; then, the absorbed nitrogen is assimilated through nitrate reductase (NR), nitrite reduction (NiR), glutamine synthetase (GS), and glutamine-2-oxoglutarate aminotransferase (GOGAT) cycle processes [6]. The metabolic products are then converted into different nitrogen-containing compounds for carbon metabolism, including photosynthesis, phytohormones, and fatty acid biosynthesis [19,20,21]. These processes can be controlled by numerous genes. Therefore, a better understanding of the underlying mechanism behind nitrogen uptake, transport, and assimilation would be helpful in providing a theoretical basis for improving NUE [22].

Foxtail millet [*Setaria italica* (L.) P. Beauv] is one of the oldest crops in the world [23], and has high drought tolerance and resistance to infertility, and has been widely cultivated in Asia for food and fodder [24,25]. It is now used as a model species for genomics and basic biological processes due to its attractive qualities, including small diploid genome (2n = 18, ~420 Mb), short life cycle, prolific seed production, and C4 photosynthesis [24,26]. It was reported that foxtail millet can perform root thickening to improve N uptake under low-nitrogen conditions [27]. Transcription profiling using RNA-Seq is a successful and widely used approach to explore molecular aspects of nutrient stress [22]. Several studies have reported that this method is widely used for investigating the mechanism of low N stress tolerance on wheat [11], rice [28], rapeseed [22], and barley [29]. A previous study has explored potential regulatory factors and functional key genes in a single foxtail millet variety in response to low nitrogen [30]. However, our previous study demonstrated that there exist significant genotypic variations among different foxtail millet varieties for NUE [31]. The molecular mechanism controlling genotypic variations between different foxtail millet varieties with contrasting low-nitrogen tolerance remains unknown. Therefore, it is important to understand the underlying regulatory mechanism of NUE in foxtail millet. In the present study, two previously selected foxtail millet varieties with contrasting NUE were used to identify the physiological and molecular mechanisms in their response to long-term low-nitrogen conditions, with the aim of determining the physiological and transcriptomic differences in foxtail millet varieties with contrasting nitrogen use efficiencies, and discovering candidate genes controlling high nitrogen use efficiency of foxtail millet, which can provide more practical and meaningful theories for improving the NUE of foxtail millet.

## 2. Results

### 2.1. Phenotypic Characteristics and N Accumulation of Foxtail Millet

Significant differences were observed between JG20 and JG22 varieties in terms of seedling growth and biomass under low-nitrogen conditions (Figure 1A,B). The shoot and root biomass of both varieties were significantly decreased under low-nitrogen conditions compared with the control, and the shoot decrease rates were 59.8% and 85.1% and the root decrease rates were 63.4% and 81.0% in JG20 and JG22, respectively (Figure 1A,B). The LN-sensitive variety JG22 exhibited higher shoot and root biomass than JG20 under a sufficient nitrogen supply, whereas the shoot and root biomass of LN-tolerant JG20 were 86.7% and 52.7% higher than that of JG22 under low-nitrogen conditions (Figure 1A,B). The N concentration and N content of the two varieties decreased significantly under low-nitrogen conditions compared with the control (Figure 2A). The N concentrations of shoots in the LN-sensitive variety JG22 were higher than in the LN-tolerant variety JG20 under low-nitrogen conditions, yet N content showed the inverse effect, where the N content in the LN-tolerant variety JG20 was 39.9% and 30.6% higher than N content in the shoot and root of the LN-sensitive variety JG22 under LN conditions (Figure 2B), which was attributed to it having a high biomass under LN (Figure 1B). Nitrogen use efficiency (NUE) varied significantly between the two varieties (Figure 2C). The NUE of shoots and roots were equal in the two varieties under normal N supply, whereas NUE in the LN-tolerant variety JG20 were 86.7% and 52.7% higher than that of JG22 in shoots and roots under low-nitrogen conditions (Figure 2C).

### 2.2. Physiological Characteristics and Response of Foxtail Millet to Low Nitrogen

Low nitrogen has no significant effects on the concentration of soluble protein except in the shoots of the LN-tolerant variety JG20, which was significantly increased by 17.6% in LN compared with the control (Figure 3A). The soluble protein concentration of shoots in the LN-tolerant variety JG20 was 29.8% higher than that in the LN-sensitive variety JG22, whereas the soluble protein concentration of the roots in JG20 was 18.3% lower than that in JG22 under low-nitrogen conditions (Figure 3A). The free amino acid concentrations of shoots and roots in JG20 and JG22 were both reduced by 21.5%, 49.8% and 41.4%, 34.8% under low nitrogen compared with the control (Figure 3B); this was caused by the synthesis of protein using amino acid to keep a normal protein concentration to maintain metabolism. A comparison of free amino acid concentration between the two varieties was consistent with the soluble protein concentration under low-nitrogen conditions; the free amino acid concentration in the LN-tolerant variety JG20 was 33.7% higher in shoots and 31.6% lower in the roots than those in the LN-sensitive variety JG22 under low-nitrogen conditions (Figure 3B). LN had different effects on the nitrate nitrogen concentration in the shoots and roots of the two varieties; the nitrate nitrogen concentration in the shoots was significantly increased by 37% and 24.8%, while in roots, it was significantly reduced by 94.0% and 90.5% in JG20 and JG22, respectively, under LN conditions compared with the control (Figure 3C).

Carbon metabolism is significantly affected by LN. The response of soluble sugar concentration to LN varied between the two varieties, LN had significantly increased soluble sugar concentration in shoots, and had no effect on soluble sugar concentration in the roots of JG20 compared with control (Figure 4A), whereas the soluble sugar concentration in the shoots was significantly reduced, and, in the roots, significantly increased in JG22 under LN conditions compared with the control (Figure 4A). The sucrose concentrations in the shoots and roots in the two varieties were increased by LN, except that, in the shoots of JG22, which was invariable between the control and LN (Figure 4B). The sucrose concentration in the shoots of JG20 was 12.1% higher than that in the shoots of JG22 under LN, while the sucrose concentration in the roots of JG20 was 40.0% lower than that in the roots of JG22 under LN (Figure 4B).

Hormones, including indole-3-acetic acid (IAA), cytokinin (zeatin), and abscisic acid (ABA), varied significantly under LN and between varieties. The indole-3-acetic acid contents of the shoots and roots were reduced by 65.0%, 39.8% and 72.4%, 66.1% in JG20 and JG22, respectively, under LN compared with the control (Figure 4C). There was no significant difference in the IAA content in the shoots between the two varieties under LN. The IAA concentration in the roots in JG22 was 2.03 fold higher than that in the roots of JG20 under LN (Figure 4C). LN had significantly reduced zeatin concentration in the shoots and roots of the two varieties, except zeatin concentration in the shoots of JG22, which was invariable between the control and LN (Figure 4D). The zeatin concentration in the shoots of JG20 was 26.4% higher than that in the shoots of JG22 under LN, while the zeatin concentration in the roots was undifferentiated between JG20 and JG22 under LN (Figure 4D). The ABA concentration was reduced in both the shoots and roots of the two varieties under LN, except the ABA concentration in the shoots of JG20, which was increased by 14.2% under LN compared with the control (Figure 4E). The ABA concentration in JG22 was 64.2% and 34.1% higher in the shoots and roots than the concentration in JG20 under LN (Figure 4E).

### 2.3. Differentially Expressed Genes (DEGs) between JG20 and JG22 under Low Nitrogen

The RNA-Seq was performed using the shoot and root samples of JG20 and JG22 seedlings, which were named SJG20-CK, RJG20-CK, SJG22-CK, and RJG22-CK for under the control, and SJG20-LN, RJG20-LN, SJG22-LN, and RJG22-LN under low-nitrogen conditions, respectively (Table 1). Using three independent biological replicates of each sample, a total of twenty-four samples were constructed and sequenced using the Solexa/Illumina platform. A total of 161 GB of raw bases were generated from the transcriptome libraries of the 24 samples, and after filtering out the adapter and low-quality sequences, approximately 150GB (95.39%) of clean reads, representing an average of 43.6 million clean reads per sample, were obtained (Table 1). On average, over 86.5% of reads were mapped to the *Setaria italic* reference genome and over 84.5% of reads were mapped to unique regions (Table 1).

To verify the RNA-Seq data, 15 DEGs regulated in response to low-nitrogen conditions were randomly selected for RT-qPCR validation in LN-tolerant variety JG20 (Figure 5A) and LN-sensitive variety JG22 (Figure 5B). The result showed that they had similar expression patterns and there was a high correlation coefficient of 0.936 (R^2^) between RNA-Seq and RT-qPCR (Figure 5C), which demonstrated that the RNA-Seq data were reliable and repeatable.

A comparison of DEGs among the varieties in response to LN conditions facilitates an understanding of the regulatory mechanisms associated with LN and helps to identify the roles of independent varieties in the regulation. Of these DEGs, a total of 4256 (JG20LN vs. JG20CK) and 1291 (JG22LN vs. JG22CK) genes in shoots and 3191 (JG20LN vs. JG20CK) and 1881 (JG22LN vs. JG22CK) genes in roots were differentially expressed in LN-tolerant JG20 and LN-sensitive JG22 foxtail millet varieties, respectively, in response to LN conditions, which showed that the LN-tolerant variety JG20 had more DEGs than the LN-sensitive variety JG22, in both shoots and roots, in response to LN conditions (Figure 6A,C, Appendix A). Meanwhile, 3581 (JG22CK vs. JG20CK) and 1132 (JG22LN vs. JG20LN) genes in shoots and 3852 (JG22CK vs. JG20CK) and 2028 (JG22LN vs. JG20LN) genes in roots were differentially expressed between the two different varieties under control and LN conditions (Figure 6A,C, Appendix A). In the combined LN-responsive and genotype-specific DEGs datasets, more than 60.7% (4362) of DEGs in the shoots and 64.8% (5133) in the roots were unique to one dataset, 35.7% (2560) of DEGs in the shoots and 32.3% (2556) in the roots were shared by two datasets, 3.4% (242) of DEGs in the shoots and 2.6% (209) in the roots were shared by three datasets, and only 0.2% (13) of DEGs in the shoots and 0.3% (20) in the roots were expressed in all four datasets (Figure 6B,D). The intersecting DEGs in JG20LN vs. JG20CK and JG22LN vs. JG22CK were 662 and 694 in the shoots and roots, respectively, among which the number of DEGs in response to LN, without considering variance in genotype, was only 494 in the shoots and 557 in the roots (Figure 6B,D).

### 2.4. Functional Categorization of DEGs in Response to Low Nitrogen in Two Foxtail Millet Varieties

The DEGs in response to low nitrogen were mainly categorized in terms of biological process (BP) and molecular function (MF) through gene ontology (GO) enrichment analysis in the two foxtail millet varieties, except for the cellular components in the shoots of the LN-tolerant variety JG20 (Figure 7). The biosynthetic and metabolic processes of carbohydrates, including the metabolic and biosynthetic processes of cellulose, polysaccharide, and glucan, were the top GO biological process (GOBP) subcategories in the shoots of JG20LN vs. JG20CK (Figure 7A). The GO subcategories of the cellular component (GOCC) were mainly focused on the extracellular region and the photosystem|, while the GO subcategories of molecular function (GOMF) mainly included transferase activity, cellulose synthase activity, and hydrolase activity in the shoots of JG20LN vs. JG20CK (Figure 7A). All GO terms contained more down- than up-regulated genes in the shoots of LN-sensitive JG22 in response to low nitrogen, where nucleoside metabolic and carbohydrate catabolic processes were the top GO subcategories for biological processes (GOBP), while the main GO subcategories of molecular function (GOMF) were phosphofructokinase activity, kinase activity, and binding in the shoots of JG22LN vs. JG22CK (Figure 7B). The gene ontology (GO) enrichment analysis showed different GO terms in the root of two foxtail millet varieties in response to low nitrogen. More down-regulated genes were enriched in the GO subcategories of biological process (GOBP), mainly focusing on nucleotide biosynthetic and metabolic processes, while transmembrane transporter activity, binding, and inhibitor activity were the main GO subcategories of molecular function (GOMF) in the roots of JG20 in response to low nitrogen (Figure 7C). The biosynthetic process and transmembrane transport of trehalose were the top GO subcategories of biological process (GOBP), while binding and transport activity were the main GO subcategories of molecular function (GOMF) in the roots of JG22LN vs. JG22CK (Figure 7D).

In shoots, a total of 680 and 236 DEGs were assigned to 119 and 94 KEGG pathways in JG20LN vs. JG20CK and JG22LN vs. JG22CK, respectively, while 529 and 397 DEGs were assigned to 111 and 107 KEGG pathways in the roots of JG20LN vs. JG20CK and JG22LN vs. JG22CK, respectively. The top 20 KEGG pathways in the four comparisons are shown in Figure 8. The significant KEGG pathways in the shoots of JG20 were related to starch and sucrose metabolism, phenylpropanoid biosynthesis, and amino acid biosynthesis and metabolism (Figure 8A), while the biosynthesis of amino acids and saccharide metabolism, including glycolysis, gluconeogenesis, and fructose and mannose metabolism were the main KEGG pathways in the shoots of JG22 (Figure 8B), which showed that DEGs of the LN-tolerant variety JG20 were not only enriched in amino acid metabolism, but also in starch and sucrose metabolism, in order to improve nitrogen use in response to low nitrogen. The KEGG pathways in roots were different from those in shoots, and varied between varieties. The DEGs in the roots of LN-tolerant JG20 were significantly assigned to carbon metabolism, transporters, and amino acid metabolism (Figure 8C) in response to LN, and plant–pathogen interactions, MAPK signaling pathways, and hormone metabolism were the main KEGG pathways in the roots of LN-sensitive JG22 under LN conditions (Figure 8D).

### 2.5. Gene Expression Profiling of Genes Involved in N Source Transport and Assimilation in Response to Low Nitrogen

The expression patterns of genes involved in N transport and assimilation in response to LN are profiled in Figure 9A–C. The nitrogen metabolism process is shown in the flowchart, and contains two parts: nitrogen uptake and transport; and nitrogen assimilation (Figure 9A). The transcript levels of genes involved in N uptake and transport varied more drastically in roots than in shoots suffering from low nitrogen in both varieties (Figure 9B). The transcription of genes NRT2.1d, NRT2.1e, NRT2.3, and NRT2.4 increased significantly in the shoots of the LN-tolerant variety JG20, whereas that in the LN-sensitive variety JG22 was invariable or slightly decreased in response to low-nitrogen conditions (Figure 9B). Compared with NRT genes in the shoots, the transcription of NRT genes was more up-regulated in roots, and varied among varieties; for example, NRT2.1a, NRT2.3, and NRT2.4 were significantly up-regulated in JG20, while NRT2.1d and NRT2.1e were more significantly up-regulated in JG22 in response to low nitrogen (Figure 9B). The expression of AMT genes varied more significantly between the two varieties than NRT genes, and the expression of AMT1.1 and AMT1.2 were both significantly increased in the shoots of the two varieties in response to LN, while AMT3.2 was uniquely up-regulated in the shoots of the LN-tolerant variety JG20 in response to LN (Figure 9B). The AMT3.1 and AMT3.2 genes showed significant increases in expression in the roots of both foxtail millet varieties under N-starved conditions, while AMT1.1 and AMT2.1 showed significantly increased expression only in the roots of JG22 in response to LN (Figure 9B).

Nitrogen assimilation was coordinated by the expression of genes involved in NR, NiR, GS, GOGAT, and GDH. The expression of NR and NiR varied significantly between the two foxtail millet varieties, which were up-regulated in both the shoots and roots of JG22, except for NiR in shoots of JG22. Meanwhile, the expressions of NR and NiR were down-regulated in the shoots of JG20, especially for NR (NADH) (Figure 9C). The expression of the genes involved in GS-GOGAT (GDH) differed among different parts and varieties. GS5, GOGAT (X3), and GOGAT2 (NADH) were significantly up-regulated, while GOGAT1 (NADH) and GDH were significantly down-regulated in the shoots of JG20 in response to LN (Figure 9C). GS1.1 showed increased expression, and GS1.3 exhibited reduced expression in the roots of both foxtail millet varieties in response to LN, which indicated that they were independent of the genotype. The expression of the gene GOGAT1 (NADH) was significantly decreased in the roots of both foxtail millet varieties, while those of GOGAT2 (NADH) and GDH (X2) were significantly up-regulated in the roots of JG20 in response to LN (Figure 9C).

### 2.6. Gene Expression Profiling of Genes Involved in Photosynthesis, Hormone Signal Transduction, and Glycolysis in Response to Low Nitrogen

Photosynthesis and glycolysis are two important pathways involved in carbon metabolism. The overall gene expression of the photosystem I (PSI), photosystem II (PS II), F-type ATPase, and light-harvesting chlorophyll protein complex (LHC) are suppressed under N limitation in the shoots of the two varieties, with the exception of genes of PsbQ and b, which are only up-regulated in the shoot of JG20 (Figure 10A, Appendix A).

Plant hormones play an important role in regulating growth and development. In the current study, the expression of genes involved in signal transduction pathways associated with several plant hormones are altered in response to LN (Figure 10B), suggesting that hormone pathways play critical roles in the shoot and root of foxtail millet in response to nitrogen availability. The genes associated with auxin-responsive protein SAUR and indole-3-acetic acid-amido synthetase GH3 are mostly down-regulated in two varieties, especially in the shoots of two varieties (Figure 10B, Appendix A), which could explain the decreased IAA content in two foxtail millet varieties under LN conditions. The downregulation of cytokinine-histidine-containing phosphotransfer protein AHP under LN is detected in roots, while it is up-regulated in the shoots of two varieties (Figure 10B, Appendix A), which could be the key gene regulating the decreased cytokinine content in shoots. The abscisic acid receptor gene PYR/PYL is up-regulated in the shoots of two varieties under LN (Figure 10B, Appendix A), this may cause increased ABA content in the shoots of two varieties under LN (Figure 10B). The other genes involved in ethylene synthesis are both down-regulated in the shoots and roots of two varieties under LN (Figure 10B, Appendix A).

Glycolysis is one of the crucial primary metabolic pathways in plants, which can supply energy and carbon skeletons for other metabolism pathways. In shoots, the downregulation of most DEGs under LN are detected, while TIM, PGK, PCK, and ALDH are up-regulated in two varieties, especially in JG20 (Figure 10C, Appendix A). AEP, ATP-PFK, PDH-E2, and DLD are up-regulated under LN in the roots of two varieties (Figure 10C, Appendix A).

### 2.7. Schematic Representation of the Main Processes and Genes Involved in Foxtail Millet in Response to Low Nitrogen

The most relevant processes and important genes related to NUE were obtained in two foxtail millet varieties (Figure 11). The nitrogen transfer genes ammonium transporters (SiAMTs) and nitrate transporters (SiNRTs) are both upregulated in the roots of two varieties, while they are only upregulated in the shoots of LN-tolerant variety JG20, and downregulated in LN-sensitive variety JG22. The nitrogen assimilation genes glutamine synthetase (SiGS) and glutamate synthase (SiGOGAT) are upregulated in the shoots of two varieties, especially in JG20 in response to LN. Nitrogen uptake and utilization efficiency are strongly associated with hormone signals; the upregulated hormone signal transduction-related genes, including auxin, cytokinine and abscisic acid, such as AUX/IAA, AFR, AHP, B-ARR, A-ARR, and PYR/PYL in shoots, and AUX/IAA, GH3, SAUR, AHP, A-ARR, PP2C, SnRK2, and ABF in roots, are found in LN-tolerant variety JG20 in response to LN. Carbon metabolism, including photosynthesis and glycolysis, significantly affects nitrogen use efficiency, which provides a carbon source and energy for nitrogen metabolism. Chlorophyll apoprotein of photosystem II and F-type ATPase genes, such as PsbQ and b, are only upregulated in LN-tolerant variety JG20 in response to LN. In the pathway of glycolysis, AEP, ATP-PFK, FBA, TIM, PGK, ALDH, ENO, and PGMP are upregulated in the shoots of LN-tolerant variety JG20 and downregulated in the shoots of LN-sensitive variety JG22 in response to LN. In the roots of two varieties, ATP-PFK are up-regulated, while the PGMP, FBA, and PCK genes are downregulated.

### 2.8. Expression Inventories of LN-Responsive TFs in Different Foxtail Millet Varieties

TFs play an important role in mediating plant growth by regulating various stress-inducible genes. In total, 144 and 37 differentially expressed TFs were detected in the shoots of JG20 and JG22, and 113 and 105 differentially expressed TFs were detected in the roots of JG20 and JG22 under low-nitrogen conditions, and these were mainly distributed among 20 TFs families (Figure 12). The number of differentially expressed TFs in the shoots of the LN-tolerant variety JG20 was higher than that in the shoots of LN-sensitive variety JG22 (Figure 12A), and TFs belonging to the WRKY (20.1%), ethylene-responsive transcription factor (EFR; 17.4%), bHLH (12.5%), and MYB (8.3%) families were the most abundant in the shoots of the LN-tolerant variety JG20, while the MYB (16.2%), bHLH (16.2%) and MADS (10.8%) families were the top TFs in the shoots of the LN-sensitive variety JG22, and the most common TFs in the shoots of the two foxtail millet varieties predominantly belonged to the bHLH (16.7%), MYB (11.1%), and MADS (11.1%) families (Figure 12A, Appendix A). The EFR and WRKY families accounted for the main differentially expressed TFs in the roots of both foxtail millet varieties that were common between them, with proportions corresponding to 19.5%, 21.9%, 21.2%, 9.7%, 18.2%, and 23.8% in JG20, JG22, and common between them, respectively (Figure 12B, Appendix A).

## 3. Discussion

Nitrogen, as a key nutrient element for plant growth and development, plays a vital role in agricultural production. Numerous studies have demonstrated that large genotypic variation in NUE exists in different crops [13,14,15,32]. Our previous research revealed significant variations in NUE among different foxtail millet varieties in our previous research [31,33]. Consistent with studies in other crops inhibited by long-term LN [34,35,36], both the shoot and root growth of foxtail millet were restrained under long-term LN conditions in the present study (Figure 1A,B), but this varied between varieties. The LN-tolerant variety JG20 had a superior shoot and root system compared to the LN-sensitive variety JG22 under long-term LN, indicating that JG20 had higher LN tolerance under long-term conditions. It has been reported that LN reduces foxtail millet growth, resulting in a considerably shorter root system and an increased root/shoot ratio [27], which was verified in the LN-sensitive variety JG22 under LN conditions in the present study (Figure 1A, Appendix A); however, the LN-tolerant variety JG20 displayed a longer root system and decreased root/shoot ratio (Figure 1A, Appendix A), indicating that the foxtail millet variety with higher nitrogen use had developed a longer root system as an obvious morphological response to N deficiency and increased the ratio of dry matter accumulation in the shoots under long-term LN; this was consistent with the findings in *Brassica napus* [37].

The N concentration and N content of two foxtail millet varieties were both significantly reduced when exposed to long-term low nitrogen (Figure 2A,B), which was consistent with our previous research in foxtail millet [31]. The N concentrations in the shoots and roots of the LN-sensitive variety JG22 was higher than in those of the LN-tolerant variety JG20 (Figure 2A), while the N contents in the shoots and roots of the LN-sensitive variety JG22 were lower than in those of the LN-tolerant variety JG20 under LN conditions (Figure 2B). This suggested that the LN-tolerant variety can efficiently utilize low N nitrogen concentrations to accumulate substantial dry matter in both shoots and roots, as indicated by the high NUE (Figure 2C). Interestingly, the soluble protein of shoots and roots in the two varieties under LN conditions remained comparable to the control, and even the soluble protein of shoots in the LN-tolerant variety JG20 increased under LN conditions (Figure 3A), which has also been observed in the early stages in maize ear and in foxtail millet root [27,38]. This suggests that foxtail millet can maintain normal levels of soluble protein to support various metabolic activities in response to LN, especially for the LN-tolerant variety. The free amino acid is an important stress indictor [39], and, in the present result, the free amino acid concentrations of the two varieties were both decreased in the shoots and roots under LN (Figure 3B), consistent with previous findings in foxtail millet [27]. The decrease in free amino acid can explain the synthesis of equivalent soluble protein from amino acid under LN conditions. The LN-tolerant variety JG20 exhibited higher concentrations of free amino acid in shoots but lower concentrations in roots compared to the control, indicating that the high-N-use foxtail millet variety can allocate more free amino acids to the shoot to ensure the N demand of photosynthetic apparatuses and other processes under low-nitrogen conditions. This observation is further supported by the significant increase in nitrate concentration, as a nitrogen source, in shoots, and the decrease in the roots of both foxtail millet varieties in response to long-term low nitrogen (Figure 3C).

Hormones play an important role in regulating plant development, growth, and adaptation to environmental stress [40]. Among them, auxin is known to regulate cell division, elongation, and differentiation during plant development and growth, and is also important in N signaling [41,42]. In this study, we found that the auxin concentration (IAA) and zeatin concentrations were significantly lower in the shoots and roots under LN (Figure 4C,D), leading to smaller shoot and root systems under LN (Figure 1A,B). This is consistent with findings in maize [38] and in foxtail millet [27]. Additionally, ABA accumulation could be an essential hormonal regulatory mechanism for N limitation, and can restrain growth in foxtail millet [27]. We observed an increase in ABA concentration in the shoots of the LN-tolerant variety JG20 and a decrease in the roots of both varieties (Figure 4E). Comparing the two varieties, the LN-sensitive variety JG22 had higher ABA concentrations in both shoots and roots compared with the LN-tolerant variety JG20, which could explain why the LN-tolerant variety was able to develop well under LN conditions.

To enhance our comprehension of the molecular mechanisms involved in the response to long-term LN in foxtail millet, we conducted a comparative transcriptional analysis between the two varieties with contrasting NUE using RNA-Seq. Our analysis identified over 10,400 differentially expressed genes (DEGs) that responded to LN in the two varieties. The analysis of expression patterns during long-term N-deficient conditions between the two varieties revealed an abundant diversity in gene expression, indicating that these stress-responsive genes may play functional roles in the N stress tolerance of foxtail millet. Notably, the LN-tolerant variety exhibited a higher number of DEGs than the LN-sensitive variety in response to long-term LN. This indicates that the LN-tolerant variety possesses a relatively stronger ability to adapt to N deficiency by activating a greater number of responsive genes. Among these DEGs, only a small number of DEGs (494 DEGs in the shoots and 557 DEGs in the roots) were shared in both varieties, disregarding the genotypic variation in response to LN. This observation highlights significant differences in the molecular genotypic variation in the response to LN among foxtail millet varieties.

The interaction between the metabolisms of C and N determines plant growth and development [43], and soluble sugars such as sucrose, glucose, and fructose, which function as an energy source in plants, play an important role in various physiological and biological processes, including growth, development, and stress response [44]. In this study, we observed a significant induction of genes encoding several enzymes involved in the metabolism of starch, sucrose, and glycolysis in response to LN in the two foxtail millet varieties. Numerous genes encoding soluble sugars, such as beta-glucosidase, glucanotransferase, and sucrose-phosphatase, were up-regulated in both the shoots and roots of the two varieties (Appendix A). Notably, the LN-tolerant variety JG20 exhibited a significantly higher number of up-regulated genes in its shoots compared to the LN-sensitive variety JG22 (Appendix A). Additionally, the trehalose 6-phosphate synthase SiTPS1 gene, a key gene involved in glycometabolism under LN conditions, showed significant up-regulation exclusively in the shoots of the LN-tolerant JG20 variety. This finding aligns with the observed increase in soluble sugar and sucrose concentration in the shoots of JG20 under LN stress (Figure 4A,B). Conversely, the ratio of up-regulated genes to total DEGs in the roots of JG22 was higher than that in the roots of JG20, which correlates with the higher soluble sugar and sucrose concentrations in the roots of JG22 compared with in JG20 (Figure 4A,B). Glycolysis can supply energy and carbon skeletons for other metabolism pathways, which is one of the important pathways of C metabolism. In this pathway, the downregulation of most DEGs in glycolysis under LN are detected in the shoots of two varieties, while aldose 1-epimerase, SiAEP and ATP-dependent 6-phosphofructokinase1, SiATP-PFK1 genes are only upregulated in LN-tolerant variety JG20 (Figure 10C, Appendix A). Most of genes of glycolysis are down-regulated in the roots of two varieties under LN, while fructose bisphosphate aldolase, SiFBA1 is significantly up-regulated under LN in root of JG20 (Figure 10C, Appendix A). These two genes play important roles in maintaining a normal metabolic level of glycolysis in JG20 under LN. It was reported that LN inhibits photosynthesis in B. napus, and varied between different genotypes [37,45]. It was reported that LN decreases photosynthesis and down-regulates the expression of genes involved in photosystem I (PSI) and photosystem II (PSII) [46]. In this research, genes encoding the components of both the light and dark reactions of photosynthesis are mostly inhibited, while only the chlorophyll apoprotein of photosystem II gene, SiPsbQ and ATP synthase subunit gene, Sib are up-regulated in LN-tolerant variety JG20 (Figure 10A, Appendix A); these should be the key genes in maintaining high photosynthesis under LN conditions.

Many studies have demonstrated that the synthesis of plant hormones is tightly regulated in response to stresses [40,47]. It has been reported that genes encoding hormones are influenced by LN stress [48]. In both varieties, the expression of genes involved in the synthesis ofindole-3-acetic acid and zeatin, such as indole-3-acetic acid-amido synthetase, auxin-responsive protein, and histidine-containing phosphotransfer protein, were down-regulated in both varieties in response to N deficiency (Figure 10B, Appendix A). This down-regulation aligns with the observed decrease in indole-3-acetic acid and zeatin concentration under LN (Figure 4C,D). Additionally, the ABA concentrations in both varieties were increased in response to LN (Figure 4E), which can be attributed to the up-regulation of abscisic acid receptor genes in foxtail millet in response to LN (Appendix A). Furthermore, our results indicate that the histidine-containing phosphotransfer peotein gene, SiAHP1 and two-component response regulator ARR-A gene, SiA-ARR are specifically up-regulated in the shoots of LN-tolerant variety JG20 under LN conditions (Figure 10B, Appendix A), resulting in higher zeatin concentration compared to the LN-sensitive variety JG22 (Figure 4D).

Nitrogen metabolism, comprising both N acquisition and transport, as well as N assimilation, was directly affected by LN conditions. Nitrate transporters and ammonium transporters are two important transport systems for nitrogen uptake [49]. It has been reported that NRT1.1 was an important gene in nitrate transport [50]; in our study, high-affinity nitrate transporters genes NRT2.3 and NRT2.4 were specifically up-regulated in the shoots and roots of the LN-tolerant variety JG20 under long-term LN conditions. Similarly, the ammonium transporter gene AMT3.2 was only up-regulated in the shoots of the LN-tolerant variety JG20 under N deficiency, while its expression in the LN-sensitive variety JG22 remained consistent or slightly decreased in response to low-nitrogen conditions (Figure 9B). This indicated that the high-affinity nitrate transporter genes NRT2.3 and NRT2.4, along with the ammonium transporter gene AMT3.2, may likely play important roles in facilitating nitrogen transport in the LN-tolerant foxtail millet variety under N deficiency; this was consistent with research in rapeseed and oil palm [51,52]. The GS/GOGAT cycle, as the main pathway of nitrogen assimilation [53,54], is considered to be an important potential maker for selecting high-NUE genotypes in wheat [55]. In our study, the genes GS5, GOGAT (X3) and GOGAT2 (NADH) were significantly up-regulated in the shoots of JG20 in response to LN (Figure 9C), indicating their crucial involvement in nitrogen assimilation in LN-tolerant foxtail millet under LN conditions. It has been found in field experiments with rice [56] that glutamate dehydrogenase (GDH) facilitates the interchange between NH4+ and glutamate, playing a minor role in nitrogen assimilation [11,57] (Figure 9A). The GDH genes in the shoots of the LN-tolerant variety JG20 were significantly down-regulated under LN, which is consistent with the findings in high-NUE wheat [11]. This showed that nitrogen assimilation in the high-NUE genotype may be more dependent on the GS/GOGAT cycle.

This indicated that LN-tolerant foxtail millet variety exhibits increased nitrogen uptake and utilization attributing to the upregulation of SiNRT2.3, SiNRT2.4 in roots and SiAMT3.2 and SiGS5 in the shoots under LN conditions. The elevated concentration of zeatin, regulated by the high expression of SiAHP1 and SiA-ARR genes, is likely an important hormonal signal for coping with LN stress in the LN-tolerant foxtail millet variety. Additionally, it is evident that enhancing the synthesis and accumulation of soluble sugars, promoting photosynthesis, and facilitating glycolysis metabolism in shoots under LN are crucial traits for achieving high nitrogen use efficiency (NUE) in foxtail millet. These traits can be regulated by key genes, such as SiTPS1, SiAEP, SiATP-PFK1, SiPsbQ, and Sib. Understanding these pathways and the associated genes should greatly contribute to genetic improvement in NUE in foxtail millet (Figure 11).

TFs, which regulates the expression of other genes in metabolic pathways [58], have been reported to be involved in the regulation of nitrogen metabolism [22,59]. Previous studies have highlighted the significance of MYB-like TFs in response to low-nitrogen stress in foxtail millet [31]. In our current study, we observed a substantial involvement of TFs in regulating metabolism in response to LN, with different expression patterns of TFs observed in both foxtail millet varieties (Figure 12A). Among these TFs, the WRKY (20.1%), ethylene-responsive transcription factor (EFR; 17.4%), bHLH (12.5%), and MYB (8.3%) families were found to be the most abundant in the shoots of the LN-tolerant variety JG20. Conversely, the MYB (16.2%), bHLH (16.2%), and MADS (10.8%) families were the predominant TFs in the shoots of the LN-sensitive variety JG22 (Figure 12A, Appendix A). These findings suggest that WRKY and EFR TFs may play vital roles in regulating the responses of foxtail millet to N deficiency for the LN-tolerant foxtail millet variety. Additionally, in both varieties, the bHLH (16.7%), MYB (11.1%), and MADS (11.1%) families were commonly expressed in shoots, while the EFR (21.2%) and WRKY (23.8%) families in roots were predominantly expressed in roots (Figure 12B). This indicates that these TFs play roles in root regulation that are independent of foxtail millet genotype.

## 4. Materials and Methods

### 4.1. Materials and Experimental Design

Two foxtail millet varieties, that were identified in our previous research, were used in this experiment. Jigu20 (JG20), a low nitrogen (LN)-tolerant variety, and Jigu22 (JG22), an LN-sensitive variety, have contrasting biomass and NUE under LN conditions. They were both bred by the crop research institute, Shandong Academy of Agricultural Science, with different parents and belong to different genotypes. The experiment was conducted in a rain-proof shed in the crop research institute, Shandong Academy of Agricultural Science, in Jinan city, in the province of Shandong, China.

The experiment was designed with a completely randomized design, with two nitrogen levels—0.2 mmol L^−1^ (low nitrogen, LN) and 6 mmol L^−1^ (normal nitrogen, CK)—and a total of four treatments: Jigu20 under normal nitrogen conditions (JG20-CK), Jigu20 under low-nitrogen conditions (JG20-LN), Jigu22 under normal nitrogen conditions (JG22-CK), and Jigu22 under low-nitrogen conditions (JG22-LN). The seeds of two foxtail millet varieties were sown on 20 June 2020 in a plastic pot with the following dimensions: 150 mm diameter × 130 mm height, with a permeable bottom. The pots were filled with 3kg purified sand substrate of 1.28 g∙cm^−3^. Each treatment has three replicate pots, each pot was sown with 100 seed density, and the seedlings were thinned to a density of 10 plants per pot according to the normal foxtail millet production density at five days after emergence. Plants were watered once with modified Hoagland nutrition [32] at 7 days after seeding with two N levels (0.2 and 6 mmol L^−1^) every 2 days. Before spraying the nutrient solution, the pots were always watered enough with purified water to wash off the residual nutrient element.

The plants were harvested at 15 days after the treatment; half of the plants in each pot were grouped in dry samples, which were separated into shoots and roots, and were dried at 105 °C for 30 min and then to a constant weight at 75 °C. The dry samples were used for weighing the biomass and determining the nitrogen content. Fresh samples of shoots and roots were immediately frozen in liquid nitrogen and then stored at −80 °C, before being taken for determination of the soluble protein concentration, free amino acid content, nitrate nitrogen concentration, soluble sugar concentration, sucrose concentration, hormone content, gene expression level, and for RNA-seq analysis.

### 4.2. Measurement of NUE

The biomass of the shoots and roots was weighed to calculate the NUE, using the following equation [60]: NUE = $\frac{Amount\;of\;absorbed\;N(shoot/root)}{Amount\;of\;supplied\;N}\times 100\%$.

### 4.3. Physiological Measurements

The N concentration of the plants was determined using the Kjeldahl method. The N content of plants was calculated using the dry matter weight multiplied by the N concentration. Soluble proteins were extracted and analyzed using a standard kit with bovine serum albumin [61]. Total free amino acid concentration was measured according to the Rose ninhydrin colorimetric method using leucine as standard [27]. Nitrate nitrogen was extracted with boiling water for 30 min, and nitrate concentration was determined using the nitrosalicylic acid colorimetric method at 410 nm [62]. Hormones including endogenous IAA, ZA, and ABA were quantified using a High-Performance liquid Chromatography (HPLC) system [63,64].

### 4.4. cDNA Library Contruction and Sequencing for RNA-Seq

cDNA library construction and sequencing for RNA-seq were based on methods described in our previous research [65]. The total RNA was extracted from samples using Trizol^®^ (Invitrogen, Carlsbad, CA, USA). The RNA Nano 6000 Assay Kit for the Bioanalyzer 2100 system (Agilent Technologies, Santa Clara, CA, USA) and a Nano Photometer spectrophotometer (IMPLEN, Westlake Village, CA, USA) were used to assess the integrity and purity of RNA, respectively. A total amount of 1 μg high-quality RNA per sample was used for the RNA sample preparations. Sequencing libraries were generated using NEBNext^®^ UltraTM RNA Library Prep Kit for Illumina^®^ (NEB, Ipswich, MA, USA), following the manufacturer’s recommendations. The sequencing of the constructed cDNA libraries was carried out at Novogene Bioinformatics Technology Co., Ltd. (Beijing, China).

Raw data of fastq format were initially processed to remove reads containing adapters, reads containing poly N, and low-quality sequence reads (>50% bases with Q-values ≤ 20), and the Q20, Q30, GC contents, and sequence duplication of the clean data were analyzed. Then, the clean reads were aligned with the foxtail millet genome (https://www.ncbi.nlm.nih.gov/genome/?term=Setaria%20italica, accessed on 14 May 2012) using HISAT2 v2.0.5 to identify relevant sequences. The mapped reads of each sample were assembled using StringTie (v1.3.3b) using a reference-based approach [66].

Gene expression levels were calculated based on FRKM (Fragments Per Kilobase of transcript sequence per Millions base pairs sequenced) values [67]. Differentially expressed genes (DEGs) between the two treatments were identified using DESeq2 R package (1.20.0). Threshold values of FDR (the false discovery rate) ≤ 0.05 and absolute log2fold-change ≥ 1 were applied to judge the significance of differences in gene expression levels [68,69].

Gene ontology (GO) and Kyoto encyclopedia of genes and genomes (KEGG) analyses for the DEGs were conducted using cluster Profiler R package. GO terms with corrected *p*-value < 0.05 were considered significantly enriched by differentially expressed genes and the top 20 KEGG pathways were selected.

### 4.5. Validation of DEGs Using qRT-PCR

To identify the reliability of the RNA-Seq results, the expression of 15 randomly selected DEGs was examined through qRT-PCR analysis using SYBR Premix Ex Taq (Clontech Takara, Shiga, Japan) on a 7500 Real-Time PCR System machine (Applied Biosystems, Foster City, CA, USA) [65]. qRT-PCR data were standardized with SiActin (gene ID: 101779009) as an internal reference. The gene-specific primes are presented in Appendix A. The gene expression of qRT-PCR was calculated using relative expression to the transcription level of SiActin in each sample using the 2^−∆∆CT^ method [70].

### 4.6. Statistical Analysis

Means and standard errors were calculated using the data from three independent samples. Analyses of variance were carried out using SPSS18.0 (SPSS 18.0, SPSS Inc., Chicago, IL, USA) and the least significant difference method (LSD) was used to test the differences for biomass, N content, NUE, and physiological parameters between the control and LN treatments of two foxtail millet varieties. Significance was specified at the level of *p* < 0.05.

## 5. Conclusions

In summary, our analyses of the physiological and transcriptomics datasets revealed that the LN-tolerant foxtail millet variety exhibits greater biomass accumulation, nitrogen content, and NUE. This can be attributed to the high metabolism capacity in its shoots as a result of having high soluble sugar, soluble protein, and zeatin concentrations, as well as low ABA concentrations, and to its having a superior root system compared with the LN-sensitive variety under LN. This is supported by numerous genes linked to high NUE in LN-tolerant foxtail millet. These genes are involved in nitrogen uptake and assimilation, photosynthesis, starch and sucrose metabolism, glycolysis, hormone metabolism, and TFs. Furthermore, we identified new candidate genes linked to LN tolerance, which could facilitate the development of foxtail millet varieties with improved NUE.

## Figures and Tables

**Figure 1 ijms-24-16321-f001:**
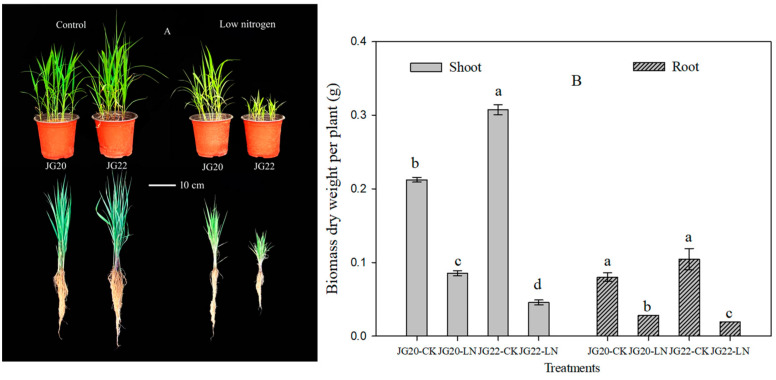
Effect of low nitrogen on phenotypic characteristics and biomass of two foxtail millet varieties with different low-nitrogen tolerance. (**A**) Phenotypic characteristics of JG20 and JG22 under control and low-nitrogen conditions; Bar = 10 cm. (**B**) Biomass of shoots and roots of JG20 and JG22 under control and low-nitrogen conditions. JG20-CK, JG20 under control; JG20-LN, JG20 under low nitrogen; JG22-CK, JG22 under control; JG22-LN, JG22 under low nitrogen; error bars represent standard error of three biological replicates; different lowercase letters indicate significance at the level of *p* < 0.05 between different treatments and varieties in shoots and roots.

**Figure 2 ijms-24-16321-f002:**
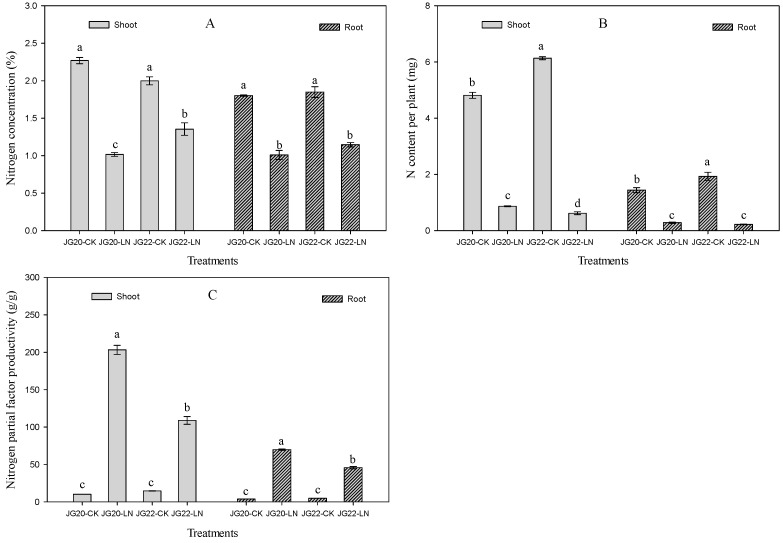
Effect of low nitrogen on N concentration, N content, and N use efficiency of two foxtail millet varieties with different low-nitrogen tolerances. (**A**) Nitrogen concentration of shoots and roots of JG20 and JG22 under control and low-nitrogen conditions. (**B**) Nitrogen content of shoots and roots of JG20 and JG22 under control and low-nitrogen conditions. (**C**) Nitrogen use efficiency of shoots and roots of JG20 and JG22 under control and low-nitrogen conditions. JG20-CK, JG20 under control; JG20-LN, JG20 under low nitrogen; JG22-CK, JG22 under control; JG22-LN, JG22 under low nitrogen. Error bars represent standard error of three biological replicates; different lowercase letters indicate significance at the level of *p* < 0.05 between different treatments and varieties in shoots and roots.

**Figure 3 ijms-24-16321-f003:**
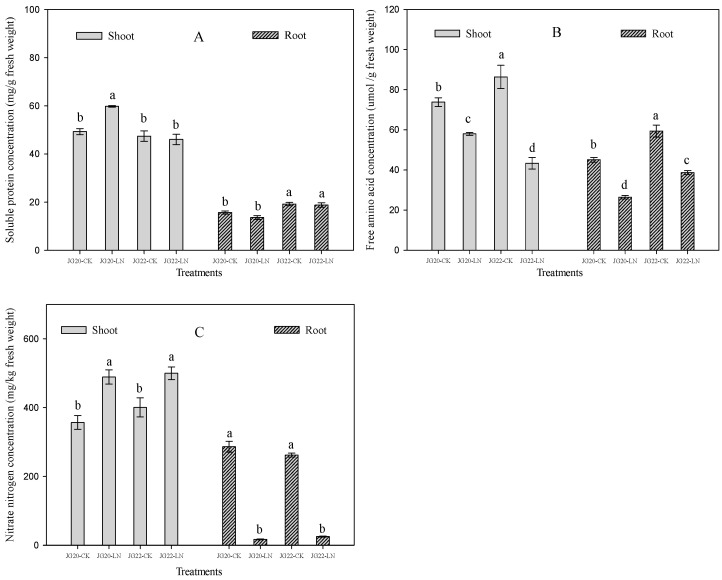
Effect of low nitrogen on soluble protein concentration, free amino acid concentration, and nitrate nitrogen concentration of two foxtail millet varieties. (**A**) Soluble protein concentration of shoots and roots of JG20 and JG22 under control and low-nitrogen conditions. (**B**) Free amino acid concentration of shoots and roots of JG20 and JG22 under control and low-nitrogen conditions. (**C**) Nitrate nitrogen concentration of shoots and roots of JG20 and JG22 under control and low-nitrogen conditions. JG20-CK, JG20 under control; JG20-LN, JG20 under low nitrogen; JG22-CK, JG22 under control; JG22-LN, JG22 under low nitrogen. Error bars represent standard error of three biological replicates; different lowercase letters indicate significance at the level of *p* < 0.05 between different treatments and varieties in shoots and roots.

**Figure 4 ijms-24-16321-f004:**
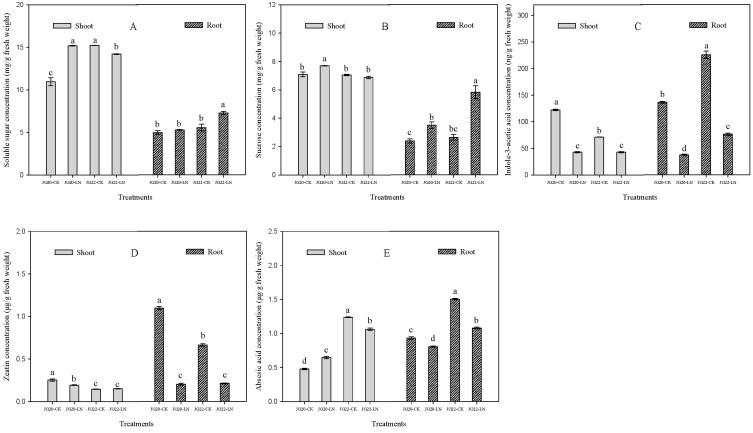
Effect of low nitrogen on soluble sugar concentration, sucrose concentration, indole-3-acetic acid concentration, zeatin concentration, and abscisic acid concentration of two foxtail millet varieties with different low-nitrogen tolerances. (**A**) Soluble sugar concentration of shoots and roots of JG20 and JG22 under control and low-nitrogen conditions. (**B**) Sucrose concentration of shoots and roots of JG20 and JG22 under control and low-nitrogen conditions. (**C**) Indole-3-acetic acid concentration of shoots and roots of JG20 and JG22 under control and low-nitrogen conditions. (**D**) Zeatin concentration of shoots and roots of JG20 and JG22 under control and low-nitrogen conditions. (**E**) Abscisic acid concentration of shoots and roots of JG20 and JG22 under control and low-nitrogen conditions. JG20-CK, JG20 under control; JG20-LN, JG20 under low nitrogen; JG22-CK, JG22 under control; JG22-LN, JG22 under low nitrogen. Error bars represent standard error of three biological replicates; different lowercase letters indicate significance at the level of *p* < 0.05 between different treatments and varieties in shoots and roots.

**Figure 5 ijms-24-16321-f005:**
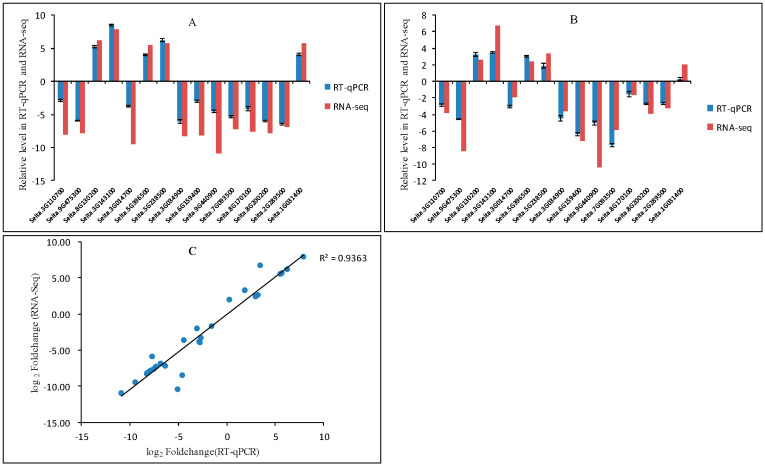
Comparison of the expression profiles of selected DEGs determined through RT-qPCR and RNA-Seq analyses. Expression levels of 15 DEGs in the low-nitrogen tolerant variety JG20 (**A**) and the low-nitrogen sensitive variety JG22 (**B**). Values are presented as log2 (fold-change). The X-axis represents gene ID, according to the NCBI database. (**C**) Scatter plots of the expression levels of 15 DEGs in control and low-nitrogen conditions. X and Y axes represent log2 (fold-change) determined through RAN-Seq and RT-qPCR, respectively (R^2^ = 0.936). Error bars represent the standard error of three biological replicates.

**Figure 6 ijms-24-16321-f006:**
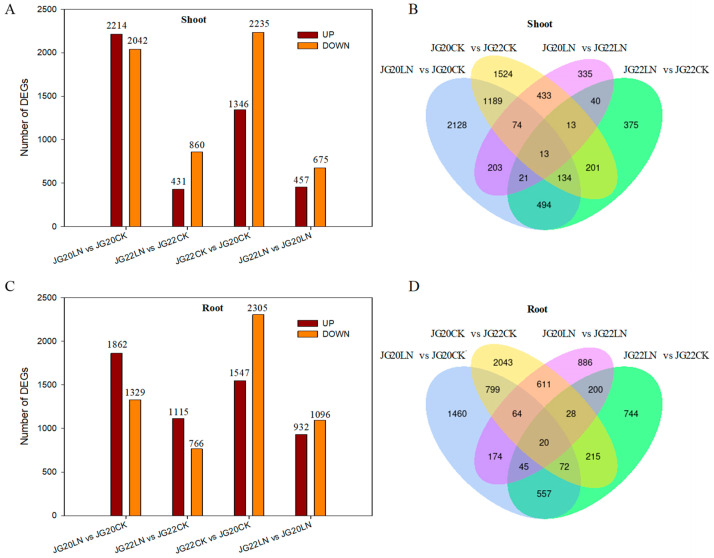
Gene expression analyses of JG20 and JG22 subjected to low-nitrogen conditions. (**A**) The numbers of DEGs in shoots in different comparison groups. (**B**) Venn diagram of the numbers of DEGs in shoots in different comparisons among groups. (**C**) The numbers of DEGs in roots in different comparison groups. (**D**) Venn diagram of the numbers of DEGs in roots in different comparisons among groups. The threshold for differential expression was set at log2 fold-change > 1 and padj < 0.05.

**Figure 7 ijms-24-16321-f007:**
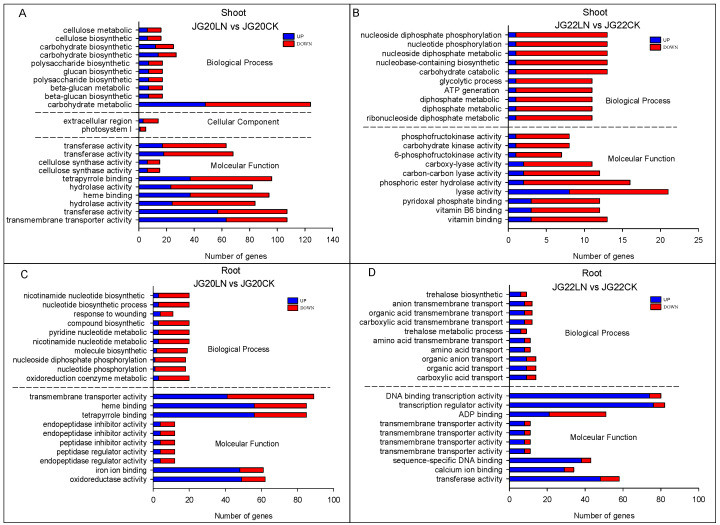
Gene ontology (GO) enrichment of DEGs in response to low nitrogen (LN) in LN-tolerant JG20 and LN-sensitive JG22 foxtail millet varieties. (**A**) GO enrichment of DEGs in the shoots between LN and CK in JG20. (**B**) GO enrichment of DEGs in the shoots between LN and CK in JG22. (**C**) GO enrichment of DEGs in roots between LN and CK in JG20. (**D**) GO enrichment of DEGs in roots between LN and CK in JG22. Blue columns indicate the numbers of up-regulated genes, while red columns indicate numbers of down-regulated genes. The threshold for differential expression was set at log2 fold-change > 1 and padj < 0.05.

**Figure 8 ijms-24-16321-f008:**
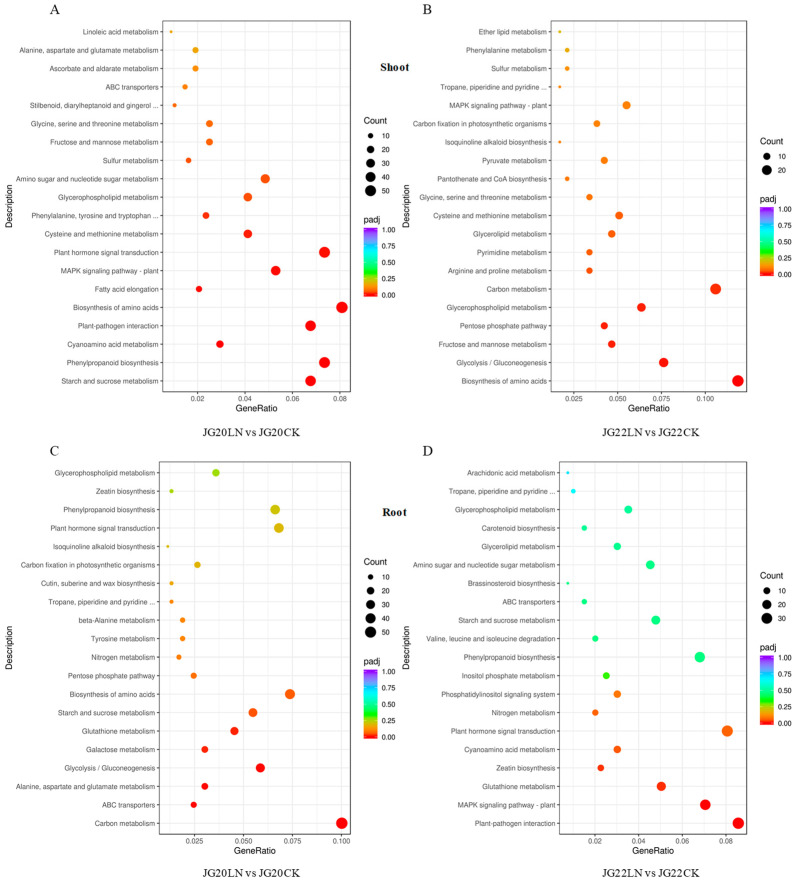
Scatter plots of top 20 KEGG pathways in two varieties in response to low nitrogen. (**A**) KEGG analysis of DEGs identified in the shoots of JG20 between CK and LN. (**B**) KEGG analysis of DEGs identified in the shoots of JG22 between CK and LN. (**C**) KEGG analysis of DEGs identified in the roots of JG22 between CK and low nitrogen LN. (**D**) KEGG analysis of DEGs identified in the roots of JG22 between CK and low nitrogen LN. Gene Ratio shows the ratio of the number of DEGs in a specific pathway to the total number of DEGs in KEGG. Pathways are listed along the y-axis. The circle area indicates the number of DEGs, and the circle color represents the ranges of the corrected *p*-values.

**Figure 9 ijms-24-16321-f009:**
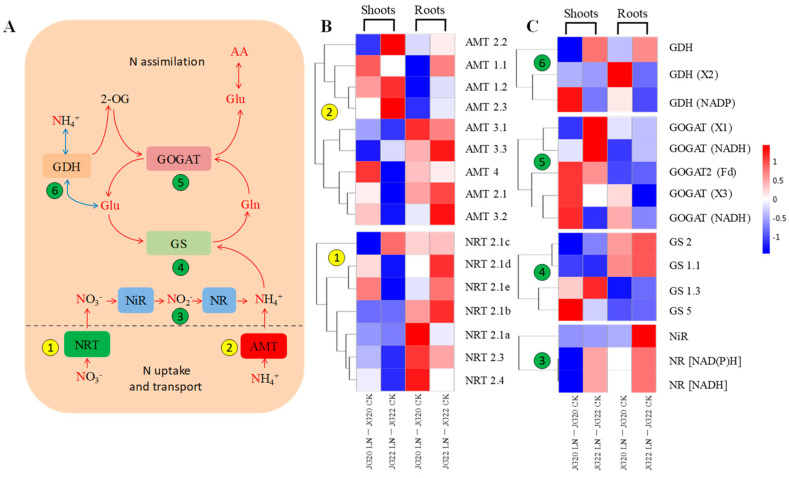
Expression profiles of genes involved in nitrogen (N) source uptake, transport, and assimilation in the shoots and roots of two foxtail millet varieties in response to low nitrogen. (**A**) Outline of N uptake, transport, and assimilation. (**B**) Heatmap visualization of expression profiles of inorganic N source transporters (NRT and AMT) in N-starved foxtail millet shoots and roots. (**C**) Heatmap visualization of expression profiles of inorganic N source assimilation (NR, NiR, GS, GOGAT, and GDH) in N-starved foxtail millet shoots and roots. Circle 1 indicates NRT (nitrate transporter); Circle 2 indicates AMT (ammonium transporter); Circle 3 indicates NR (nitrate reductase) and NiR (nitrite reductase); Circle 4 indicates GS (glutamine synthetase); Circle 5 indicates GOGAT (glutamate synthetase); Circle 6 indicates GDH (glutamate dehydrogenase); Glu, glutamate; Gln, glutamine; 2-OG, 2-oxoglutarate; AA, amino acid. The bar on the right side of the heatmap represents relative expression level of DEGs. The threshold for differential expression was set at log2 fold-change > 1 and padj < 0.05.

**Figure 10 ijms-24-16321-f010:**
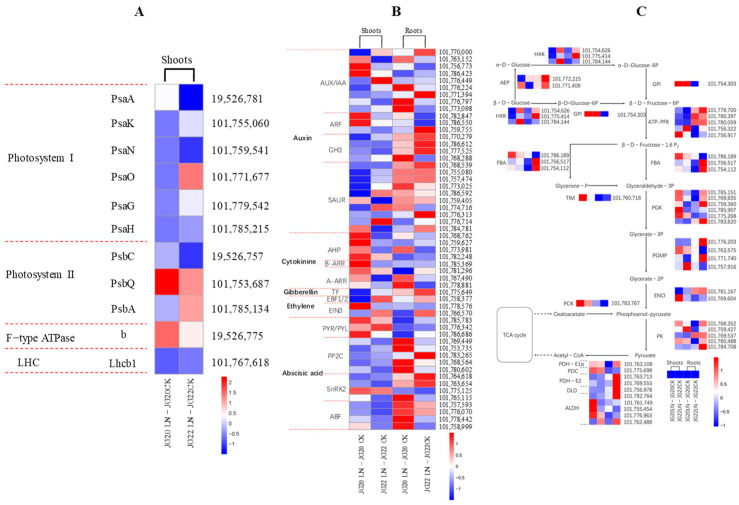
Heatmap visualization of expression profiles of genes involved in photosynthesis (**A**), hormone signal transduction (**B**), and glycolysis (**C**) in two foxtail millet varieties in response to low nitrogen. Psa, chlorophyll apoprotein of photosystem I; Psb, chlorophyll apoprotein of photosystem II; b, ATP synthase subunit b; LHC, light-harvesting chlorophyll protein complex; Lhcb1, light-harvesting complex II chlorophyll b binding protein; AUX, Auxin; IAA, Indole-3-acetic acid; AFR, Auxin response factor; GH3, indole-3-acetic acid-amido synthetase GH3; SAUR, auxin-responsive protein SAUR; AHP, histidine-containing phosphotransfer protein; A-ARR/B-ARR, two-component response regulator ORR; TF, transcription factor; EBF, EIN3-binding F-box protein; EIN3, ethylene-insensitive protein 3; PYR/PYL, abscisic acid receptor; PP2C, probable protein phosphatase 2C; SnRK2, serine/threonine-protein kinase; ABF, G-box-binding factor; HXK, hexokinase; AEP, aldose 1-epimerase; GPI, glucose-6-phosphate isomerase; ATP-PFK, ATP-dependent 6-phosphofructokinase; FBA, fructose-bisphosphate aldolase; TIM, triosephosphate isomerase; PGK, phosphoglycerate kinase; PGMP, phosphoglycerate mutase-like protein; ENO, enolase; PCK, phosphoenolpyruvate carboxykinase; PK, pyruvate kinase; PDH-E1α, pyruvate dehydrogenase E1 component subunit alpha; PDC, pyruvate decarboxylase; PDH-E2, pyruvate dehydrogenase E2 component; DLD, dihydrolipoamide dehydrogenase; ALDH, aldehyde dehydrogenase. The threshold for differential expression was set at log2 fold-change > 1 and padj < 0.05.

**Figure 11 ijms-24-16321-f011:**
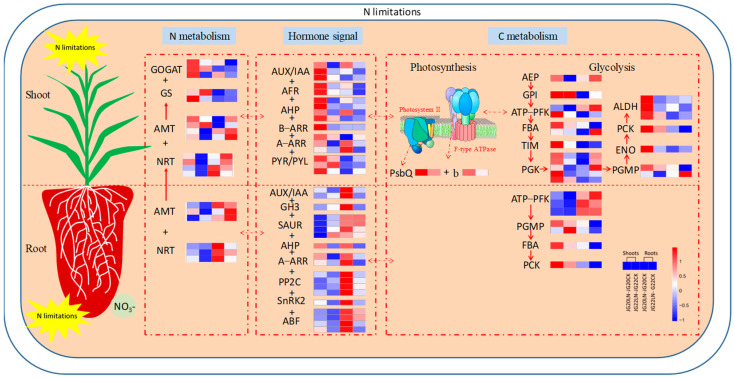
Schematic representation of the main processes and important genes involved in low nitrogen response in two foxtail millet varieties with different nitrogen use efficiencies. The color scale represents increased (red) or decreased (blue) fold-change expression of DEGs in samples exposed to low nitrogen. NRT, nitrate transporter; AMT, ammonium transporter; GS, glutamine synthetase; GOGAT, glutamate synthetase; AUX, Auxin; IAA, Indole-3-acetic acid; AFR, Auxin response factor; GH3, indole-3-acetic acid-amido synthetase GH3; SAUR, aux-in-responsive protein SAUR; AHP, histidine-containing phosphotransfer protein; A-ARR/B-ARR, two-component response regulator ORR; PYR/PYL, abscisic acid receptor; PP2C, probable protein phosphatase 2C; SnRK2, serine/threonine-protein kinase; ABF, G-box-binding factor; Psb, chlorophyll apoprotein of photosystem II; b, ATP synthase subunit b; AEP, aldose 1-epimerase; GPI, glucose-6-phosphate isomerase; ATP-PFK, ATP-dependent6-phosphofructokinase; FBA, fructose-bisphosphate aldolase; TIM, tri-osephosphate isomerase; PGK, phosphoglycerate kinase; PGMP, phosphoglycerate mutase-like protein; ENO, enolase; PCK, phosphoenolpyruvate carboxykinase; ALDH, aldehyde dehydrogenase. The threshold for differential expression was set at log2 fold-change > 1 and padj < 0.05.

**Figure 12 ijms-24-16321-f012:**
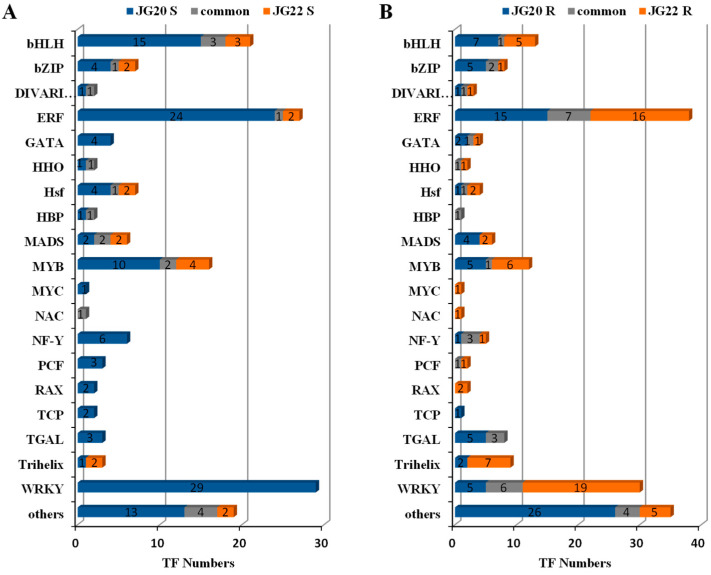
Changes in transcription factor (TF) expression in different foxtail millet varieties. (**A**) Distribution of TF families in the shoots of JG20 and JG22 in response to low nitrogen. (**B**) Distribution of TF families in the roots of JG20 and JG22 in response to low nitrogen.

**Table 1 ijms-24-16321-t001:** Quality statistics of read numbers of foxtail millet transcriptomics.

Sample Name	Raw Reads	Clean Reads	Clean Bases	Error Rate (20%)	Q20 (%)	Q30 (%)	GC Content (%)	Total Mapped	Uniquely Mapped	Multiple Mapped
SJG20-CK	45,434,210	44,021,351	6.60 G	0.02	98.18	94.78	56.54	42,466,340 (96.47%)	40,838,306 (92.77%)	1,628,034 (3.70%)
RJG20-CK	46,926,017	44,908,641	6.73 G	0.02	98.32	95.10	53.21	38,087,242 (84.81%)	37,391,122 (83.26)	696,119 (1.55%)
SJG20-LN	46,568,083	45,266,295	6.79 G	0.02	98.15	94.69	56.37	43,654,878 (96.44%)	42,879,943 (94.73%)	774,935 (1.71%)
RJG20-LN	45,728,815	44,559,130	6.68 G	0.02	98.11	94.52	52.95	33,272,477 (74.67%)	32,684,085 (73.35%)	588,392 (1.32%)
SJG22-CK	42,193,254	41,110,815	6.17 G	0.02	98.36	95.08	53.98	39,846,088 (96.92%)	38,835,094 (94.46%)	1,010,994 (2.46%)
RJG22-CK	43,185,923	42,196,855	6.33 G	0.02	98.33	94.95	50.22	30,066,139 (71.25%)	29,460,001 (69.82%)	606,138 (1.44%)
SJG22-LN	44,891,345	43,960,698	6.59 G	0.02	98.13	94.55	54.73	42,392,060 (96.43%)	41,541,056 (94.50%)	851,004 (1.94%)
RJG22-LN	43,616,102	42,607,394	6.39 G	0.02	98.24	94.77	52.21	31,884,267 (74.83%)	31,199,409 (73.23%)	684,858 (1.61%)

Notes: SJG20-CK, Shoot of JG20 under control; RJG20-CK, Root of JG20 under control; SJG20-LN, Shoot of JG20 under low nitrogen; RJG20-LN, Root of JG20 under low nitrogen; SJG22-CK, Shoot of JG22 under control; RJG22-CK, Root of JG22 under control; SJG22-LN, Shoot of JG22 under low nitrogen; RJG22-LN, Root of JG22 under low nitrogen; Raw Reads: The number of reads before filtering; Clean Reads: Filtered reads; Total Clean Bases (Gb): Total number of bases after filtration; Q20 (%): Proportion of nucleotides with a quality value larger than 20 in the filtered reads; Q30 (%): Proportion of nucleotides with a quality value larger than 30 in the filtered reads; GC content: Percent of GC bases on total bases.

## Data Availability

The datasets generated and analyzed during the current study are available in the National Center for Biotechnology Information (NCBI) Sequence Read Archive (SRA) database (http://www.ncbi.nlm.nih.gov/sra/, accessed on 14 May 2022) (Accession number: PRJNA838221). Additional supporting tables are included as Appendix A. All plant materials are available from the Crop Research Institute, Shandong Academy of Agricultural Sciences, P. R. China.

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
