# Peer review of "Physiological and Transcriptomic Analysis Provides Insights into Low Nitrogen Stress in Foxtail Millet (Setaria italica L.)"

_ijms, 2023, doi:10.3390/ijms242216321_

Round 1

Reviewer 1 Report (New Reviewer)

Comments and Suggestions for Authors

The submitted manuscript is topical in terms of the use of genetic analyses and understanding the relationships between N uptake, translocation and utilization in plants. The conclusions presented can be applied to other crops. The question is whether they can be generated when there are physiological differences between monocotyledonous plants (to which the species under study belonged) and dicotyledonous plants. This could be noted in the summary. I have a few comments on the introductory section. At the beginning, the issue of nitrogen and its relationship to the environment (water pollution, leakage into the atmosphere, etc.) is addressed. I would recommend that this section is also addressed, as there is, for example, a so-called Nitrates Directive within the EU. I believe this is necessary with regard to citing publications, this way it only has reach within Asia. Furthermore, I recommend to better formulate the objectives or hypotheses of the work. The results and their description rely on their graphical representation. It would be useful to improve the quality of the graphs, especially the axis descriptions, which are harder to read. The description of the results does not indicate whether the differences are conclusive or inconclusive. The description that something is higher/lower is insufficient. Values need to be added to the results section as the authors are dealing with specific results, in some cases the use of relative values (%) is justified. The discussion is descriptive in places. The authors should also focus on confronting their results with others and draw some conclusions. Are all literature sources of older date necessary?

Author Response

Dear Reviewer,

Thank you for your prompt and thorough review of our manuscript, " Physiological and Transcriptomic Analysis to Provides Insights into Low Nitrogen Stress in Foxtail Millet (Setaria italica (L.) P. Beauv).).," which we submitted to the International Journal of Molecular Sciences. We appreciate the valuable feedback we received from the reviewer, and we have made significant revisions to our manuscript to address their concerns.

We have carefully considered all of the comments made by the reviewers and have taken steps to improve the quality and clarity of our manuscript. Furthermore, we have addressed all formatting, language errors and citation issues as per the journal guidelines, and have made sure that all references are up-to-date and accurately cited.

We have worked diligently to ensure that our revised manuscript meets the high standards of the International Journal of Molecular Sciences, and we are confident that our work will make a valuable contribution to the field of research.

Thank you again for your time and consideration. We look forward to hearing from you soon.

Sincerely,

Erying Chen

Reviewer 1 Comments

Comments 1. The submitted manuscript is topical in terms of the use of genetic analyses and understanding the relationships between N uptake, translocation and utilization in plants.

The conclusions presented can be applied to other crops.

The question is whether they can be generated when there are physiological differences between monocotyledonous plants (to which the species under study belonged) and dicotyledonous plants. This could be noted in the summary.

Response: Thank you for your comments. In our study, we have concluded that the findings are specific to foxtail millet, as outlined in our conclusions (lines 771-773). While similar conclusions have been drawn for other monocotyledonous crops, such as maize (line 525) and wheat (line 626), the need for further investigation remains, especially in the case of dicotyledonous plants.

Comments 2. I have a few comments on the introductory section. At the beginning, the issue of nitrogen and its relationship to the environment (water pollution, leakage into the atmosphere, etc.) is addressed. I would recommend that this section is also addressed, as there is, for example, a so-called Nitrates Directive within the EU. I believe this is necessary with regard to citing publications, this way it only has reach within Asia.

Response: We have revised the introductory section and added nitrates directive within the EU in lines 51-52 to increase the range of application of this article.

Comments 3. Furthermore, I recommend to better formulate the objectives or hypotheses of the work.

Response: We have rewritten the purpose to make the aim of article more specific and logical, this was revised in lines 87-90.

Comments 4. The results and their description rely on their graphical representation. It would be useful to improve the quality of the graphs, especially the axis descriptions, which are harder to read.

Response: we have improved the quality of the graphs (Fig 5, Fig7, Fig 9 and Fig12) and revised or added the axis instruction, and described the axis in fig instruction to make it more clearly (Fig 1, Fig2, Fig 3 and Fig 4).

Comments 5. The description of the results does not indicate whether the differences are conclusive or inconclusive. The description that something is higher/lower is insufficient. Values need to be added to the results section as the authors are dealing with specific results, in some cases the use of relative values (%) is justified.

Response: We have revised the description of the results, and added the relative values to make the description of result more specific, which can be tracked in the result.

Comments 6. The discussion is descriptive in places. The authors should also focus on confronting their results with others and draw some conclusions.

Response: We have revised the description of the discussion, and confronted our result with other studies, and also made some conclusions, which can make the discussion more descriptive

Comments 7. Are all literature sources of older date necessary?

Response: We have deleted the unnecessary references, such as previous 3 and 5

Reviewer 2 Report (New Reviewer)

Comments and Suggestions for Authors

General comments:

The manuscript is robust but still several points need to be addressed. English might need to be improved.

Detailed comments:

The title needs to be revised. 

“Integration of Physiological and Transcriptomic Analysis…” or

“An Integrated Physiological and Transcriptomic Approach…” might be more functional.  

The integrated approach is very limited. It can be advisable to include analyses which are able to “integrate” the physiological and molecular results. PCA could be a solution to estimate the variation of genes and the physiological behaviour to them related. Please consider an example of this approach (Pagano et al., 2022; doi: 10.1021/acsnano.1c08367)

Figures: 

Figure 5, please avoid the Gene ID, or include in the description the DEGs function or gene name.

Figure 7 need to be improved: molecular function is poorly informative and sometimes redundant. It is better to give more relevance to biological processes, which might give more info on the pathways involved, and, if interested to understand the organelles involved the cellular component.

Figure 9 might need to implement a gene clustering, which can be obtained easily with R.

Figure 12 is very difficult to follow. Moreover, data are not scalar. And the percentage may increase the complexity of the figure.

Comments on the Quality of English Language

English might need to be improved.

Author Response

Dear Reviewer,

Thank you for your prompt and thorough review of our manuscript, " Physiological and Transcriptomic Analysis to Provides Insights into Low Nitrogen Stress in Foxtail Millet (Setaria italica (L.) P. Beauv).).," which we submitted to the International Journal of Molecular Sciences. We appreciate the valuable feedback we received from the reviewer, and we have made significant revisions to our manuscript to address their concerns.

We have carefully considered all of the comments made by the reviewer and have taken steps to improve the quality and clarity of our manuscript. Furthermore, we have addressed all formatting, language errors and citation issues as per the journal guidelines, and have made sure that all references are up-to-date and accurately cited.

We have worked diligently to ensure that our revised manuscript meets the high standards of the International Journal of Molecular Sciences, and we are confident that our work will make a valuable contribution to the field of research.

Thank you again for your time and consideration. We look forward to hearing from you soon.

Sincerely,

Erying Chen

General comments:

The manuscript is robust but still several points need to be addressed. English might need to be improved.

Detailed comments:

The title needs to be revised. 

“Integration of Physiological and Transcriptomic Analysis…” or

“An Integrated Physiological and Transcriptomic Approach…” might be more functional.  

The integrated approach is very limited. It can be advisable to include analyses which are able to “integrate” the physiological and molecular results. PCA could be a solution to estimate the variation of genes and the physiological behaviour to them related. Please consider an example of this approach (Pagano et al., 2022; doi: 10.1021/acsnano.1c08367)

Response: The previous title aimed to utilize a Physiological and Transcriptomic Approach in order to ascertain the physiological and transcriptomic differences between two foxtail millet varieties exhibiting contrasting nitrogen use efficiencies. Based on our findings, it was determined that the suggested Principal Component Analysis (PCA) may not be appropriate for this article. Consequently, we have revised the title to "Physiological and Transcriptomic Analysis Provides Insights into Low Nitrogen Stress in Foxtail Millet (Setaria italica L.)" to better reflect the content of the study.

Figures: 

Comments 1.

Figure 5, please avoid the Gene ID, or include in the description the DEGs function or gene name.

Response: We have changed the Gene ID to gene name

Comments 2.

Figure 7 need to be improved: molecular function is poorly informative and sometimes redundant. It is better to give more relevance to biological processes, which might give more info on the pathways involved, and, if interested to understand the organelles involved the cellular component.

Response: We have simplified and specified the molecular function and biological processes in Figure7, which may be easier to understand now

Comments3.

Figure 9 might need to implement a gene clustering, which can be obtained easily with R.

Response: We have added the gene clustering in the Figure9

Comments4.

Figure 12 is very difficult to follow. Moreover, data are not scalar. And the percentage may increase the complexity of the figure.

Response: We have changed the percentage to numbers, and make new scalar in Figure 12

Comments5.

Comments on the Quality of English Language English might need to be improved.

Response: We have improved the quality of English language to make it more clearly.

Round 2

Reviewer 2 Report (New Reviewer)

Comments and Suggestions for Authors

the manuscript has been improved in all its parts.

This manuscript is a resubmission of an earlier submission. The following is a list of the peer review reports and author responses from that submission.

Round 1

Reviewer 1 Report

Comments and Suggestions for Authors

Commentos to author

- Authors haver not yet introduce what statistical analysis they made.
- There are abbreviation problems. CK is cytokinin and authors use it to
describe the normal nitrogen condition which is confusing.
- Which statistical test was used?
- Are they really no error bars for JG20LN and JG22LN Figure 1? For a field
experiment I really doubt it.
- I understand the reasoning of the article and the results. Lots of analysis were conducted. However, I fail to see how authors interconnect them in the
discussion. In the introduction it is mentioned: “Our results reveal the
molecular mechanism of foxtail millet in response to chronic low nitrogen and identifies genes associated with high low nitrogen (LN) tolerance, which could assist in the genetic improvement of NUE in foxtail millet.”. However, nothing related to this is discussed in the discussion section. Authors should provide some genes which should be used for this, or at least suggest which
metabolic pathway is more interesting to look forward as a tool for breeding.

Author Response

Dear reviewer

Re: Manuscript ID: ijms- 2412538 and Title: “Physiological and Transcriptomic Analysis Provides Insights into Low Nitrogen Stress in Foxtail Millet (Setaria italica L.).”

Thanks very much for you comments concerning our manuscript entitled “Physiological and Transcriptomic Analysis Provides Insights into Low Nitrogen Stress in Foxtail Millet (Setaria italica L.).” (ijms-2412538). Those comments are valuable and very helpful. We have read through comments carefully and have made corrections. Hope these will make it more acceptable for publication. Revisions in the text are tracked. The responses to the comments are presented as following.

Response to Reviewer 1 Comments

Point 1: - Authors have r not yet introduce what statistical analysis they made.

Response 1: The statistical analysis is described in 4.6 Statistical analysis (lines 716-721)

Point 2:- There are abbreviation problems. CK is cytokinin and authors use it to describe the normal nitrogen condition which is confusing.

Response 2: We have searched the whole manuscript and not found for the abbreviation CK for cytokinin, and we only used CTK for cytokinin in the Conclusion Section (line 727)

Point 3: - Which statistical test was used?
Response 3: The least significant difference method (LSD) was used to test the significance of differences, which is described in line 719.

Point 4: - Are they really no error bars for JG20LN and JG22LN Figure 1? For a field
experiment I really doubt it.

Response 4: JG20LN and JG22 LN do have standard errors but the values are too small, and due to the shrinking of the Figure 1, it seems that there are no error bars for JG20LN and JG22LN in Figure 1. Now we supply the raw data and standard error for Figure 1 in supplemental excel. The experiment was conducted in pot.

Point 5:- I understand the reasoning of the article and the results. Lots of analysis were conducted. However, I fail to see how authors interconnect them in the discussion. In the introduction it is mentioned: “Our results reveal the molecular mechanism of foxtail millet in response to chronic low nitrogen and identifies genes associated with high low nitrogen (LN) tolerance, which could assist in the genetic improvement of NUE in foxtail millet.”. However, nothing related to this is discussed in the discussion section. Authors should provide some genes which should be used for this, or at least suggest which metabolic pathway is more interesting to look forward as a tool for breeding.

Response 5: According to the suggestion of reviewer, we have revised the introduction and discussion, we added and discuss the pathway and related important genes which could assist in the genetic improvement of NUE in foxtail millet, other already discussed genes are also marked in red (lines 552-619).

Reviewer 2 Report

Comments and Suggestions for Authors

The presented manuscript is devoted to the study of the effect of low N content on the Foxtail Millet. The work was done at a high level, combining biochemical, genetic and bioinformatic methods. The N assimilation scheme also adds value to the study. However, the manuscript is not without some remarks.

1 - page numbering after Figure 5 has changed

39- rephrase the sentence. DEGs cannot be enriched with BP and MF

147 - rephrase the sentence. Pathway cannot participate in carbohydrate metabolism.

Rice. 2 - concentration is the content of something in something. mg in what? N/plant content in one plant?

In the caption to Fig. 2, there is a gap between efficiency and of

Fig. 9 The signature does not contain the designations of numbers

The Discussion section provides the rationale and objectives for this study, which can be moved to the Introduction section.

434-10% AcOH does not hydrolyze proteins and peptides to free aa, even 10% TCA is used to isolate proteins. Ninhydrin stains the entire amine content of the molecule

443 - The purity and integrity of RNA on a spectrophotometer is difficult to determine, only by electrophoresis and chromatography.

Author Response

Dear reviewer

Re: Manuscript ID: ijms- 2412538 and Title: “Physiological and Transcriptomic Analysis Provides Insights into Low Nitrogen Stress in Foxtail Millet (Setaria italica L.).”

Thanks very much for your comments concerning our manuscript entitled “Physiological and Transcriptomic Analysis Provides Insights into Low Nitrogen Stress in Foxtail Millet (Setaria italica L.).” (ijms-2412538). Those comments are valuable and very helpful. We have read through comments carefully and have made corrections. Hope these will make it more acceptable for publication. Revisions in the text are tracked. The responses to the comments are presented as following.

Response to Reviewer 2 Comments

The presented manuscript is devoted to the study of the effect of low N content on the Foxtail Millet. The work was done at a high level, combining biochemical, genetic and bioinformatic methods. The N assimilation scheme also adds value to the study. However, the manuscript is not without some remarks.

Point 1: 1 - page numbering after Figure 5 has changed

Response 1: we have fixed the problem with successive numbering after Figure 5

Point 2: 39- rephrase the sentence. DEGs cannot be enriched with BP and MF

Response 2: we have rephrased the sentence (now line 254)

Point 3: 147 - rephrase the sentence. Pathway cannot participate in carbohydrate metabolism.

Response 3: we have rephrased the sentence (now line 363)

Point 4: Rice. 2 - concentration is the content of something in something. mg in what? N/plant content in one plant?

Response 4: we solved the problem by remaking the diagram with correct concentration unit in Figure 2

Point 5: In the caption to Fig. 2, there is a gap between efficiency and of

Response 5 : we have added the gap between efficiency and of (line 119)

Point 6: Fig. 9 The signature does not contain the designations of numbers

Response 6 : we have added the explanation of the numbers into the caption of the diagram in Fig.9

Point 7: The Discussion section provides the rationale and objectives for this study, which can be moved to the Introduction section.

Response 7 : According to the suggestion of reviewer, we have revised the introduction and discussion, and integrated part of them (lines 76-90).

Point 8: 434-10% AcOH does not hydrolyze proteins and peptides to free aa, even 10% TCA is used to isolate proteins. Ninhydrin stains the entire amine content of the molecule

Response 8 : The original expression of the method described above was inaccurate. To avoid the misunderstanding, we have rephrased the description of the method (lines 673-675).

Point 9: 443 - The purity and integrity of RNA on a spectrophotometer is difficult to determine, only by electrophoresis and chromatography.

Response 9 : The 2100 Bioanalyzer system is a mature automated electrophoresis tool for biomolecular sample quality control. The system provides timely digital data for objective evaluation of molecular weight, quantification, integrity and purity of DNA, RNA and protein, which has been widely used in previous research, such as in

1)https://doi:10.1111/pbi.13033; 2)https://doi.org/10.1016/j.chom.2019.02.003;

3) https://doi.org/10.1186/s13059-018-1412-6

Round 2

Reviewer 1 Report

Comments and Suggestions for Authors

I have no more suggestions for the authors.

I failed to see though why they used LSD instead of Tukey or another more powerful post-hoc comparison test.

It's true that authors now provide more genes that could be suited for improving N uptake in foxtail millet, but that's something already known for other species.

Finally, the problem with CK is that the authors use it for normal nitrogen when that abbreviation is used worldwide for Cytokinins. It makes the reader think that the CK treatment has cytokinins.